# DIVERSITY EMPOWERS INTELLIGENCE: INTEGRATING EXPERTISE OF SOFTWARE ENGINEERING AGENTS

**Kexun Zhang**[1,2], **Weiran Yao**[1], **Zuxin Liu**[1], **Yihao Feng**[1], **Zhiwei Liu**[1], **Rithesh Murthy**[1], **Tian Lan**[1], **Lei Li**[2], **Renze Lou**[1], **Jiacheng Xu**[1], **Bo Pang**[1], **Yingbo Zhou**[1], **Shelby Heinecke**[1], **Silvio Savarese**[1], **Huan Wang**[1], **Caiming Xiong**[1]

[1]Salesforce AI Research, [2]Carnegie Mellon University

## ABSTRACT

Large language model (LLM) agents have shown great potential in solving real-world software engineering (SWE) problems. The most advanced open-source SWE agent can resolve over 27% of real GitHub issues in SWE-Bench Lite. However, these sophisticated agent frameworks exhibit varying strengths, excelling in certain tasks while underperforming in others. To fully harness the diversity of these agents, we propose **DEI** (Diversity Empowered Intelligence), a framework that leverages their unique expertise. DEI functions as a meta-module atop existing SWE agent frameworks, managing agent collectives for enhanced problem-solving. Experimental results show that a DEI-guided committee of agents is able to surpass the best individual agent's performance by a large margin. For instance, a group of open-source SWE agents, with a maximum individual resolve rate of 27.3% on SWE-Bench Lite, can achieve a 34.3% resolve rate with DEI, making a 25% improvement and beating most closed-source solutions. Our best-performing group excels with a 55% resolve rate, *securing the highest ranking* on SWE-Bench Lite. Our findings contribute to the growing body of research on collaborative AI systems and their potential to solve complex software engineering challenges[1].

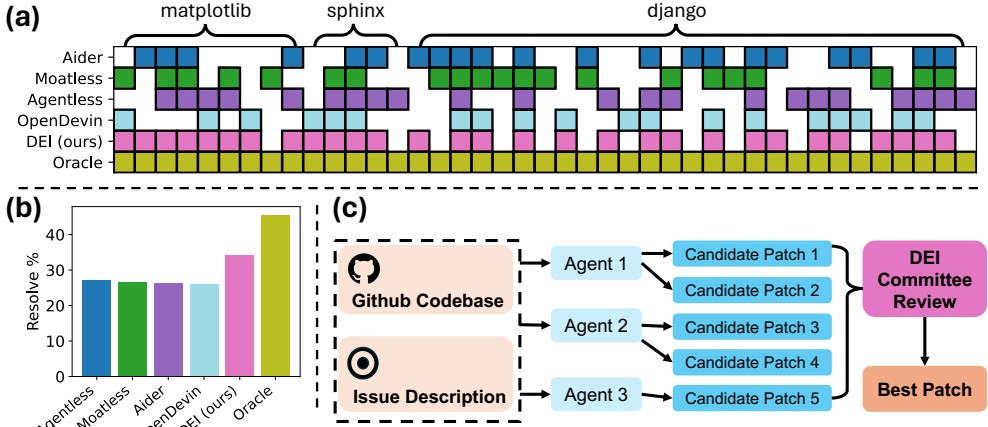

Figure 1: Different SWE agents (Aider, Moatless, Agentless, OpenDevin) resolve very different sets of issues (the colored girds in Fig **1a**), despite having similar resolve rates (Fig **1b**). Our proposed DEI Committee takes candidates patches and tries to select the best, oracle choice (Fig **1c**), improving the resolve rate significantly to be better than any single agent in the committee.

---

[1]Code, data, and generations are released at https://github.com/SalesforceAIResearch/swecomm.

# 1 INTRODUCTION

Recent advancements in large language models (LLMs) have transformed software engineering (SWE) and other domains. Originally developed as chatbots (Schulman et al., 2022; OpenAI-Team, 2024), LLMs have evolved into the core of AI agents, capable of understanding and generating human-like conversations, as well as autonomously executing actions in both real-world and digital environments. SWE agents, a specialized subset of these AI agents, integrate these capabilities with software engineering tools and techniques for tasks like code generation, automated testing, and project management, aiming to identify and resolve practical software issues (Zhang et al., 2024).

In this paper, we study one specific task of SWE agents – resolving real-world GitHub issues based on their descriptions. Automatically fixing a bug in a code repository is an extremely challenging task that involves navigating extensive codebases, understanding complex function interactions, detecting subtle errors, and generating the correct fix patch. The large action space of SWE agents, together with long trajectories, inevitably result in the diversity of Github issue solutions, as shown in Figure 1. We have observed that different SWE agents resolve very different sets of issues (the colored girds in Figure 1**a**), despite having similar resolve rates (Figure 1**b**). This is probably due to different skill sets of SWE agents. For instance, OpenDevin (Wang et al., 2024c) explicitly instructs the LLM to first replicate the bug in an issue and executes its replication in a development workspace to provide feedback for its generated patches, but other agents like Moatless Tools (Örwall, 2024) and Agentless (Xia et al., 2024) do not actually execute code in the issue-specific repository.

> *A garden's beauty never lies in one flower. Diversity in all its forms is the path to greatness.*

Similarly, the trend in the SWE agent community reflects this diversity—no single agent framework dominates in all capabilities. It is the flourishing variety within this community that sparks new ideas and leads to the development of better agents.

The variety in SWE agent capabilities inspires us to develop DEI, **D**iversity **E**mpowered **I**ntelligence, a framework that leverages the strengths of diverse agents. DEI aims to harness these varied skills to tackle a broader range of problems more effectively with a multi-agent ensemble system and a re-ranking pipeline, as showcased in Figure 1**c**. DEI functions as a meta-module that can be integrated with any existing agent framework, enabling scalable management and collaboration among agents to form a more powerful multi-agent software engineering organization.

We evaluate DEI on 7 groups of candidate agents on SWE-Bench Lite. 3 of the 7 are different runs of a single open-source SWE agent. The other 4 are different agents that are on the SWE-Bench Lite leaderboard, including a group that contains only open-source agents. Through experiments, we find that different agents show a great level of diversity in the issues they resolve: a group of agents with an average resolve rate of 26.6% can collectively solve 54.3% of issues if there is an oracle reviewer that can consistently select the best candidate. DEI, as a first step towards harnessing such diversity, can improve the group's resolve rate to 34.3% ($\uparrow 25\%$), suggesting that LLMs are great code reviewers. These findings mirror the benefits of diversity in the tech industry, where diverse perspectives and skills lead to greater innovation and problem-solving capabilities.

To summarize, our contributions are the following:

- For the first time, we comprehensively evaluate the diversity of solutions provided by SWE agents, revealing significant differences in the types of GitHub issues resolved by various agents, despite similar overall resolve rates. These findings suggest a substantial potential to improve overall performance by effectively leveraging the diverse expertise of these agents together.
- This paper introduces DEI, a multi-agent meta-policy module designed to harness the diversity of SWE agents and seamlessly facilitate collaboration among agents with different specialties. By employing a multi-stage rating and re-ranking pipeline, DEI consistently improves issue resolution, demonstrating a 25% performance boost on the SWE-Bench Lite leaderboard.

# 2 RELATED WORK

We review the work in fundamental language agent architecture, recent developments for SWE agents, and multi-agent or ensemble methods in this section.

**Fundamental Language Agent**  Pioneering AI agent methods along this line of work include ReAct (Yao et al., 2023), Reflexion (Shinn et al., 2023), CodeAct (Wang et al., 2024b), etc., in which ReAct interprets the user query, generates functional API calls, and gets the tool outputs in real time; Reflexion further appends failed trial experience into the memory, enabling effective retrials to prevent repetitive errors. CodeAct (Wang et al., 2024b), instead of generating function calls, uses code generation to consolidate AI agents' actions into a unified action space.

**Software Engineering Agent**  We present the SWE agents which have disclosed the technical details on the SWE-bench Lite leaderboard. Alibaba Lingma Agent (Ma et al., 2024) constructs a repository knowledge graph to represent code and dependencies, using a Monte Carlo tree search-based strategy for repository exploration, and generates patches to address real-world GitHub issues. AutoCodeRover (Zhang et al., 2024) adds advanced code search tools, such as abstract syntax trees, and spectrum-based fault localization to the agent for enhancing context understanding and issue resolution. Code-R (Chen et al., 2024) chooses a multi-agent framework with pre-defined task graphs to resolve Github issues. Agentless (Xia et al., 2024), is a simplified two-phase approach for solving software development problems. It focused on localization and repair without relying on LLMs to make decisions, highlighting the potential of straightforward techniques in autonomous software development. OpenDevin (Wang et al., 2024c) is a hub of community-contributed agents including CodeAct (Wang et al., 2024b), browser agent, GPTSwarm (Zhuge et al., 2024), and task-specific micro agents. Finally, SWE-agent (Yang et al., 2024) developed agent-computer interface that consists of LM-friendly commands and environment feedback to enable LM agents to autonomously use computers to solve software engineering tasks.

**Multi and Ensemble Agents**  Recent works observe that organizing multiple specialized AI agents (Hong et al., 2024; Li et al., 2023; Liu et al., 2024) enable the task decomposition ability of an agent system, which improves the task-resolving performance. Current multi-agent frameworks are categorized into three types based on their execution patterns. Firstly, static agent working flow (Wu et al., 2024; Github, 2023), which pre-defines the agent execution flows and ignites agent transitions via specified conditions. Controlling a multi-agent system with pre-determined states is robust, though losing flexibility in terms of unseen states or conditions. Secondly, ensemble via self-consistency (Wang et al., 2023), LLM-as-a-judge (Zheng et al., 2023), group chatting (Wu et al., 2023; Hong et al., 2024; Wang et al., 2024a; Chen et al., 2023). This is built upon an environment where multiple agents send messages to each other in a group channel such that their thoughts are ensembled. Variants of group chatting includes debating (Liang et al., 2023; Chan et al., 2023) and model-wise ensembling (Wang et al., 2024a). Last but not least, hierarchical task assignment (Liu et al., 2024; 2023). Organizing multi-agent in a hierarchical structure benefits the top-down task decomposition and thus enables efficient multi-agent collaboration.

## 3 INTEGRATING EXPERTISE OF SWE AGENTS

### 3.1 BACKGROUND

**Resolving issues in SWE-Bench.** One important task in software engineering is to resolve issues raised by developers. SWE-Bench curates instances of this task by collecting successfully resolved issues from open-source repositories on Github. Each instance in SWE-Bench consists of a textual issue description, a version of the repo just before the issue was resolved, and (hidden) unit tests that went from fail to pass after the human-written patch. To resolve an instance, the model is required to generate a patch that can pass these unit tests.

**SWE Agents.** In this paper, we use the term "SWE agents"[1] to refer to *any* LLM-based system that generates patches to solve issues in a code base, e.g., an instance in SWE-Bench. While the specific implementation varies, a typical SWE agent usually gives their underlying LLM several tools in the form of callable functions to navigate through the code base, find relevant context, edit files, and run tests. The workflow of SWE agents involves multiple LLM calls, each taking some or all outputs from previous steps as input.

---

[1]According to our definition, SWE-agent (Yang et al., 2024) is an instance of SWE agents, and Agentless (Xia et al., 2024), despite the name, is another.

## 3.2 DIVERSITY OF SWE AGENTS

We consider two types of diversity: *intra-agent diversity* and *inter-agent diversity*.

*Intra-agent diversity* is defined as the degree to which different runs of the same agent solve different problem instances. It is most likely from the non-determinism of the underlying LLM due to sampling in decoding and mixture-of-experts architecture (Chann, 2023). Since the workflow of SWE agents involves multiple steps and LLM calls, a slight difference in an earlier step can easily propagate and result in significant differences in the final outcome.

*Inter-agent diversity* is defined as the degree to which different agents solve different problem instances. Besides sharing the potential causes of intra-agent diversity, inter-agent diversity is also largely because of differences in agent design, including different tools, workflows, and prompts.

## 3.3 APPROACH

### 3.3.1 SWE AGENT PROBLEM FORMULATION

We formulate the SWE agent problem under the *contextual Markov decision process* (CMDP) framework (Hallak et al., 2015), represented by the tuple $\mathcal{M} = (\mathcal{S}, \mathcal{C}, \mathcal{A}, \mathcal{R}, \mathcal{P}, p_0, \rho)$. Here, $\mathcal{S}$ denotes the state space, which encompasses all possible states the agent could encounter, such as the current status of files. The context space, $\mathcal{C}$, includes relevant repository information and issue descriptions. The action space, $\mathcal{A}$, represents all potential actions or tools the SWE agent can utilize, such as `search` or `editing`. The context-dependent reward function, $\mathcal{R} : \mathcal{S} \times \mathcal{A} \times \mathcal{C} \to \mathbb{R}$, assigns scores based on the actions taken by the agent. For instance, the reward is high if the agent successfully addresses an issue, while it is low if the action results in new bugs in the repository. The context-dependent transition function, $\mathcal{P} : \mathcal{S} \times \mathcal{A} \times \mathcal{C} \to \Delta(\mathcal{S})$, defines how the state of the repository or information changes following a specific action. The context-dependent initial state distribution is denoted by $p_0 : \mathcal{C} \to \Delta(\mathcal{S})$, and $\rho \in \Delta(\mathcal{C})$ represents the context distribution.

Given the initial context $c \sim \rho$ and initial state $s_0 \sim p_0(\cdot \mid c)$, at each time step $t$, the agent follows a policy $\pi : \mathcal{S} \times \mathcal{C} \to \Delta(\mathcal{A})$ to select an action $a_t \sim \pi(s_t, c)$ and receives a reward $\mathcal{R}(s_t, a_t, c)$. The environment then transitions to the next state $s_{t+1} \sim \mathcal{P}(\cdot \mid s_t, a_t, c)$, providing the agent with a new state observation. As the iteration progresses to time $T$, a sampled trajectory $\tau := \{s_t, a_t, r_t\}_{t=0}^{T}$ is obtained. The objective of an SWE agent is to maximize the cumulative reward along the trajectory, which is captured by the value function:

$$\max_{\pi} V^{\pi}(\rho) = \max_{\pi} \mathbb{E}_{\tau} \left[ \sum_{t=0}^{T} \mathcal{R}(s_t, a_t, c) \mid c \sim \rho; \pi \right] \tag{1}$$

### 3.3.2 OUR FRAMEWORK: DIVERSITY EMPOWERED INTELLIGENCE (DEI)

Many efforts have been made to implement sophisticated agent systems that aim to achieve the objective described in Eq. 1. However, as discussed in Section 1, these systems often exhibit varying levels of effectiveness across different contexts. It is challenging to devise a single agent that can consistently perform well across all possible contexts. Formally, suppose there are $N$ agent policies, denoted as $\{\pi_1, \pi_2, \ldots, \pi_N\}$, where each policy is tailored to address a specific context $\{\rho_1, \rho_2, \ldots, \rho_N\}$. The union of these contexts is a subset of the entire context space, i.e., $\rho_1 \cup \rho_2 \cup \cdots \cup \rho_N \subseteq \rho$. For each agent policy $\pi_i$, the objective is:

$$\pi_i = \max_{\pi} \mathbb{E}_{\tau} \left[ \sum_{t=0}^{T} \mathcal{R}(s_t, a_t, c) \mid c \sim \rho_i; \pi \right]. \tag{2}$$

However, an agent policy $\pi_i$ may perform poorly in a different context $\rho_j$ (where $j \neq i$). To address this limitation, we propose our framework: Diversity Empowered Intelligence (DEI). The DEI framework leverages the strengths of each agent in their respective contexts to enhance overall performance across all contexts.

We introduce a meta-policy, denoted as $\pi_{\text{DEI}}$, which aims to optimally select among the available agent policies based on the context. The goal of $\pi_{\text{DEI}}$ is defined as:

$$\pi_{\text{DEI}} = \max_{\pi} \mathbb{E}_{c \sim \rho} \left[ \mathbb{E}_{\tau} \left[ \sum_{t=0}^{T} \mathcal{R}(s_t, a_t, c) \mid c; \pi(c) \right] \right], \tag{3}$$

where $\pi(c)$ denotes the selection of the optimal agent policy from $\{\pi_1, \pi_2, \ldots, \pi_N\}$ based on the observed context $c$. By dynamically choosing the most suitable agent policy for each context, the DEI framework seeks to maximize the expected cumulative reward across all possible contexts.

### 3.3.3 DEIBASE: A SIMPLE YET EFFECTIVE IMPLEMENTATION

We present **DEIBASE**, a simple yet powerful implementation of the DEI framework, tailored for SWE-Bench like problems. The context in the setup includes the repository, along with relevant files and issue descriptions. The meta-policy's action space consists of the final patches generated by different agent frameworks, each specialized in addressing various aspects of the problem.

**DEIBASE** utilizes a Large Language Model (LLM) as a code review committee. The LLM evaluates candidate patches by analyzing the state of the code base before and after the proposed changes, in conjunction with the contextual information from the issue descriptions. It produces detailed explanations for each patch, justifying the modifications based on the identified issues, the context, and the specific changes made.

While other methods of code review and scoring, such as rule-based approaches, can be incorporated into our framework, the use of an LLM-based committee offers a unique advantage. LLMs often excel at evaluating solutions when evaluation is easier than generation. **DEIBASE** thus serves as an effective baseline for LLM-based SWE evaluation, highlighting potential performance variations among diverse SWE agents and showcasing the capabilities of our method.

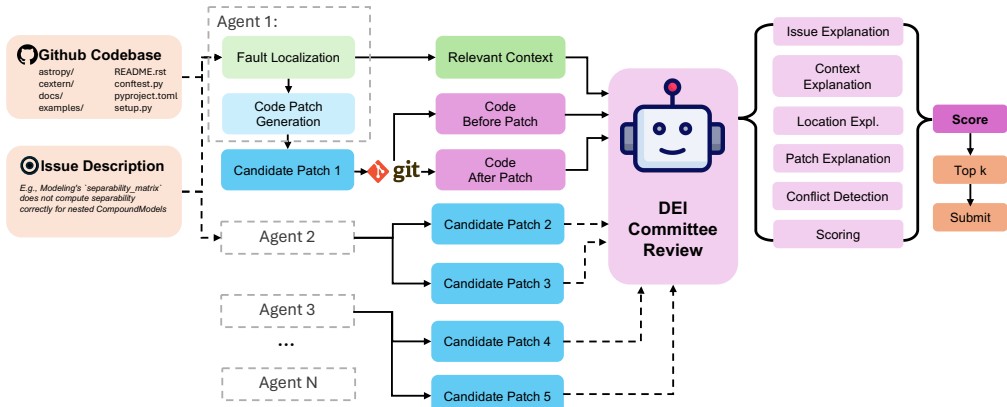

Figure 2: **Framework Overview.** DEI first examines the code base before and after a candidate patch, along with other relevant contexts. Then, it generates an explanation for the issue, the context, and the patch and tries to justify the patch. With its own explanation, it scores the candidate patches and picks the top-scoring ones as more likely to be correct.

As demonstrated in Figure 2, **DEIBASE** is given multiple candidate patches for a single issue. These patches might be from running a single SWE agent multiple times or running multiple SWE agents. **DEIBASE** gives each candidate patch a score and then selects the top-scoring candidates as the patches most likely to work. Prompts and example inputs/outputs can be found in Appendix A.5.

**Step 1: Input Construction.** Four inputs are given to **DEIBASE** for each patch: the issue description itself, relevant context (code snippets identified by an SWE agent as relevant to the issue), code before the patch, and code after the patch. The form of inputs reflects two design choices. First, the entire repository is often too large to fit directly in the context limit of LLMs, so we use the relevant context instead to save token costs and help the model focus. Second, the format of a patch is not the easiest for an LLM to read as it switches back and forth between the pre-change code and

the changed code, so we give the code before and after the patch separately to the model for easier understanding. In practice, we directly use the relevant code spans identified by Moatless Tools, an open-source SWE Agent (Örwall, 2024). There might be potential ways of improving the quality of relevant code spans by making them specific to both the issue and the candidate patch, rather than solely dependent on the issue itself.

**Step 2: Explanation Generation.** To help the model better "understand" the patch before scoring, we instruct it to generate various explanations regarding the patch in a specified order. The order is decided so that the earlier explanations can also help the later ones. We describe each explanation in the order they are generated here: 1) *Issue explanation* explains what the issue is about and what problem it may be causing. 2) *Context explanation* explains how and why each relevant code span (there might be many of these) is relevant to the issue. 3) *Location explanation* explains if and why the patch is modifying the correct part of the code that's faulty. 4) *Patch explanation* explains if and how the patch is fixing the issue. 5) *Conflict detection* is about checking whether the patch conflicts with other relevant code snippets. We explicitly instruct the model to refer back to the earlier explanations while generating the later ones.

**Step 3: Patch Scoring.** Based on its own explanations, the model is asked to give the candidate patch a score of 1 to 10. We give the model detailed rubrics of what violations/mistakes lead to higher score deductions and what should only be considered minor violations. For example, if the model finds the modification location to be wrong, it is considered a serious mistake. We score each candidate multiple times and average the score. To encourage diverse outputs, we use an inference temperature of 1.2

## 4 EXPERIMENTS

We aim to answer two research questions with our experiments: 1) How diverse are LLM-based SWE agents in terms of intra- and inter-agent diversity? 2) To what extent can DEI harness the diversity and increase the performances of these SWE agents?

### 4.1 EXPERIMENT SETUP

#### 4.1.1 BENCHMARK AND AGENTS

**Benchmark**. We conduct our experiments on SWE-Bench Lite, a 300-instance subset sampled from the full SWE-Bench for providing a more self-contained evaluation of functional bug fixes (Jimenez et al., 2024). Compared to the full SWE-Bench, SWE-Bench Lite has significantly more submissions on the leaderboard for us to conduct a more comprehensive analysis of inter-agent diversity.

**Agents.** For intra-agent diversity, we consider three well performing open-source agents on the SWE-Bench Lite leaderboard: Agentless (Xia et al., 2024), Moatless (Örwall, 2024), and Aider (Gauthier, 2024) by running them 10 times with the same parameters. For inter-agent diversity, we consider 10 agents that have similar resolve rates, all between 26.0% and 31.0% on the leaderboard by directly using their submitted patches to the SWE-Bench issues. For the evaluation of **DEIBASE** on different agents, we consider 3 groups of agents that are submitted to SWE-Bench Lite before August 15, 2024, including one group consisting of only open-source agents. For the evaluation of **DEIBASE** on multiple runs of a single agent, we use the generations of the three aforementioned agents – Agentless, Moatless Tools, and Aider. More details can be found in Appendix A.4.

#### 4.1.2 EVALUATION METRICS

We use the same set of metrics for both intra- and inter-agent diversity as these metrics are defined for multiple candidate solutions without requiring them to come from the same candidate. We assume there are $n$ candidates in total, and we are computing "@k" metrics over all of its subsets with $k$ candidates.

**Resolve rate** measures how good a SWE agent is. It is defined as the percentage of issues resolved by the agent. We measure both single SWE agents and DEI with it to see how much DEI helps.

**Union@k** measures the **best case** performance of $k$ candidates, assuming an **oracle reranker** to always pick the best candidate. For any set of size $k$, if an issue is resolved by one of the $k$ candidates,

it is counted as solved for Union@$k$. We compute the average of resolve rates over all possible sets of size $k$.

**Intersect@k** measures the **worst case** performance of $k$ candidates, assuming an **adversarial reranker** that always picks the worst candidate. An issue is considered resolved if only all $k$ candidates resolve it.

**Average@k** measures the **average case** performance by computing the average number of problems solved by $k$ candidates, assuming a **random reranker** that picks a random candidate.

**DEI@k** measures the performance of **our method** by computing the number of problems solved by the top-1 candidate deemed by DEI. The better our reranking mechanism is at telling good solutions from bad ones, the higher DEI@k is.

Our research questions can be answered by the gaps between these metrics. **Union@k - Intersect@** measures how diverse the agents are, while **DEI@k - Average@k** measures how much DEI helps in selecting the correct candidate. Note that when there are more than $k$ candidates, we consider all possible subsets of size $k$ when computing the "@k" benchmarks. We report the mean and standard deviation of the benchmarks over all subsets.

## 4.2 MAIN RESULTS

### 4.2.1 RESEARCH QUESTION 1: HOW DIVERSE ARE LLM-BASED SWE AGENTS?

To answer this question, we report the "@k" metrics of 10 different agents and 10 runs of single agents in Figure 3.

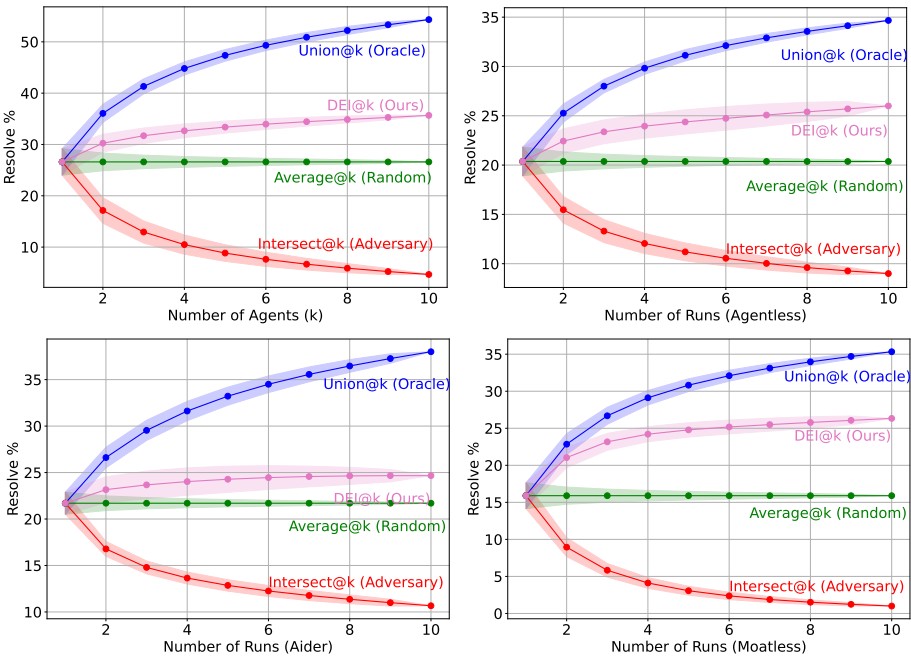

Figure 3: How different metrics change as more candidate solutions are involved. In all 4 scenarios, there is a huge gap between Union@k and Average@k. The concrete dots are expectation over all possible subsets of size $k$, while the shaded area corresponds to the standard deviation.

Several observations can be made about the results:

**SWE agents resolve very different sets of issues across agents and agent runs. Their full potential is far from fully released.** In all four subfigures, the gap between Union@k and Average@k, as well as between Average@k and Intersect@k, is large. As $k$ – the number of candidates – gets larger, the gap also gets larger. For 2 of the 4 settings, Union@k is more than 2x larger than Aver-

age@k for $k = 10$. The other 2, Union@k is more than 1.5x larger than Average@k for $k = 10$. This indicates that current SWE agents are potentially capable of resolving a lot more issues, if we have a reranker that can tell which candidates are correct.

**Different agents resolve more distinct issues than different runs of a single agent. In other words, diversity does empower intelligence.** The absolute/relative difference between Union@k and Average@k is much larger in the first subfigure than in the following three subfigures. For the "10 different agents" setting, as $k$ approaches 10, the distinct issues resolved are $2\times$ the average number of issues resolved by a single agent in the group.

Table 1: Resolve rates of top submissions on SWE-Bench Lite (cutoff date: Aug 15, 2024). We evaluate 3 DEI Committees formed by different groups of agents. Each DEI Committee outperforms the best agent in it significantly. **DEIBASE**-Open, formed by 4 open-source agents can beat many closed-source agents.

| **DEI** Group | % Resolve | System | Open Src | Trajs | Open Candidates | Backend LLM |
|---|---|---|---|---|---|---|
| **1** | **55.0** | **DEIBASE**-1 | ✓ | ✓ | ✗ | gpt4o |
| 1 | 50.6 | Cosine Genie | ✗ | ✗ | - | "Fine-tuned OpenAI" |
| 1 | 43.0 | CodeStory Aide | ✗ | ✗ | - | gpt4o, Claude 3.5 Sonnet |
| - | 38.0 | AbenteAI MentatBot | ✗ | ✗ | - | gpt4o |
| **2** | **37** | **DEIBASE**-2 | ✓ | ✓ | ✗ | gpt4o |
| **Open** | **34.3** | **DEIBASE**-Open | ✓ | ✓ | ✓ | gpt4o |
| - | 34.0 | Bytedance MarsCode | ✗ | ✗ | - | gpt4o |
| - | 33.0 | Alibaba Lingma | ✗[1] | ✗ | - | gpt-4-1106-preview |
| 2 | 31.3 | Factory Code Droid | ✗ | ✗ | - | "Anthropic and OpenAI" |
| 2 | 30.6 | AutoCodeRover | ✗[2] | ✗ | - | gpt4o |
| 2 | 29.6 | Amazon Q Dev. | ✗ | ✗ | - | Unknown |
| 2 | 28.3 | CodeR | ✗[1] | ✗ | - | gpt-4-1106-preview |
| 2 | 28.0 | MASAI | ✗[1] | ✗ | - | Unknown |
| - | 27.6 | SIMA | ✗[1] | ✓ | ✓[3] | gpt4o |
| Open | 27.3 | Agentless | ✓ | ✓ | - | gpt4o |
| Open | 26.6 | Moatless Tools | ✓ | ✓ | - | Claude 3.5 Sonnet |
| - | 26.6 | IBM Research Agent | ✗ | ✗ | - | Unknown |
| Open | 26.3 | Aider | ✓ | ✗ | - | gpt4o, Claude 3 Opus |
| Open | 26.0 | OpenDevin + CodeAct | ✓ | ✓ | - | gpt4o |

[1] Their repo has no code yet.
[2] An earlier version is open-source. The current one is not.
[3] Candidates are generated by a "modification of moatless tools".

### 4.2.2 RESEARCH QUESTION 2: HOW MUCH DOES DEI HELP?

We apply **DEIBASE** to the candidates in Figure 3 as they are added to the group. Our findings are:

**DEIBASE helps in most cases.** For most values of $k$ in all subfigures, we observe a significant improvement of DEI@k over Average@k, indicating that **DEIBASE** selects correct candidates much better than a random baseline.

**DEIBASE helps more when the candidates come from different agents.** This finding resonates with a similar finding from research question one: Since candidates from multiple agents have a larger potential for improvement (Union@k - Average@k), the actual improvements created by **DEIBASE** (DEI@k - Average@k) are also larger. This suggests that given a limited budget of candidates, it would be better to choose a diversity of agents over multiple runs of the same agent.

**As $k$ gets larger, DEIBASE's improvement first increases and then plateaus.** While larger $k$ generally indicates higher DEI@k, the margin gets smaller and there are cases when an increase in $k$ results in a slight drop in performance. This suggests that the current **DEIBASE** is not ideal for a large group of agents and there is still room for a better reranking mechanism.

Based on the lessons above, we propose three **DEIBASE** groups in which each candidate is from a different agent and no more than 5 candidates exist for each instance. The members of these **DEIBASE** groups and their performance are reported in Table 1. **DEIBASE**-1 consists of the top 2 agents. **DEIBASE**-2 consists of 5 closed-source agents that have high performance on the leaderboard. **DEIBASE**-Open consists of 4 open-source agents so that we know future researchers can run the entire pipeline. As Table 1 shows, all three **DEIBASE** instances outperform the best candidate in the group. Surprisingly, **DEIBASE**-Open shows a 7% increase in resolve rates and beats most of the closed-source systems.

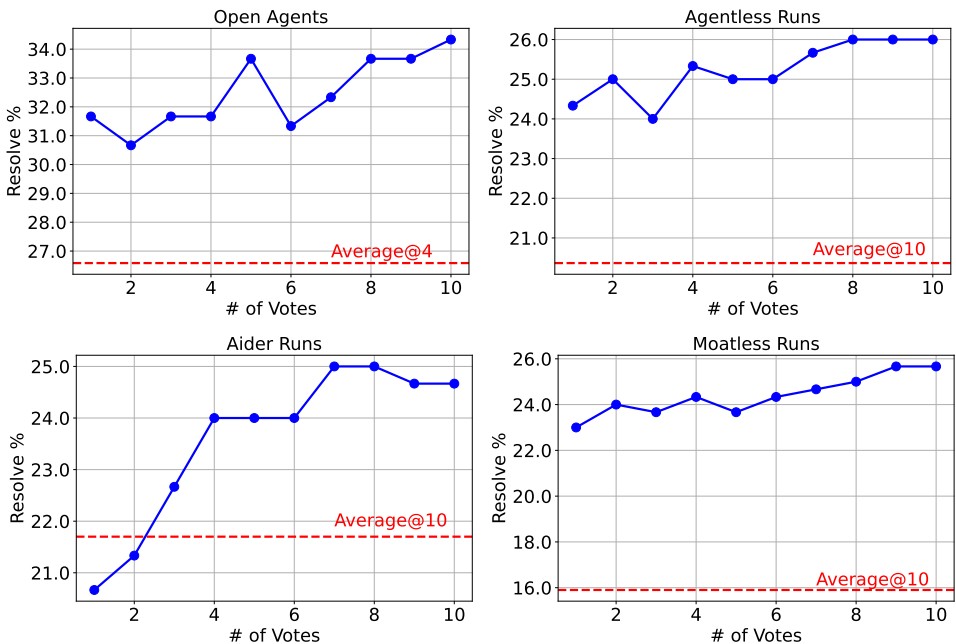

Figure 4: How the performance of **DEIBASE** changes as the LLM is given more votes for scoring.

## 4.3 ABLATION AND ANALYSES

In this subsection, we demonstrate some ablation studies to investigate the effectiveness of different components in the framework, in order to answer the following questions. To advocate for open science, all the ablation experiments are conducted on either our own reproduction of open-source SWE agents or their official generations.

**Question 1: Does DEI get better with more votes?**

**Answer 1: Yes.** Arguably, DEI itself has the same potential characteristics as SWE agents that may cause diverse outputs. However, unlike SWE agents whose outputs are patches, DEI's output is an integer score, which can easily be aggregated and averaged. This is why we give DEI more votes and rerank the candidates according to the average score. In most **DEIBASE** experiments, we allow 10 votes for each candidate patch. To investigate whether more votes lead to better patch reviewing, we directly take the scores generated for **DEIBASE**-Open, **DEIBASE**-Agentless, **DEIBASE**-Aider, and **DEIBASE**-Moatless, and check for various values of $m$, how the first $m$ scores can help us find the best patch.

As demonstrated in Figure 4, more votes generally lead to better resolve rates. Another finding is that for 3 out of the 4 evaluation settings, **DEIBASE** was able to get much better performance than the average candidate with only one vote. Even when **DEIBASE** wasn't able to get better than average with one vote, it managed to get an improvement with only three votes. These results suggest that **DEIBASE** itself also produces diverse outputs, but it is easier to aggregate them via score averaging.

**Question 2: Are the explanations necessary?**

**Answer 2: Yes.** We remove the part about asking for explanations from the prompt and compare **DEIBASE**-Open, **DEIBASE**-Agentless, **DEIBASE**-Aider, and **DEIBASE**-Moatless under the same evaluation setting with and without explanations. We report their resolve rates in Table 2. For all 4 settings we evaluated, **DEIBASE** with explanations performs slightly better than **DEIBASE** without explanations.

Table 2: Comparing **DEIBASE**'s resolve rates with and without explanations.

|  | Open Agents | Agentless | Aider | Moatless |
|---|---|---|---|---|
| **DEIBASE** w/ expl. | 34.6 | 26.0 | 24.6 | 25.6 |
| **DEIBASE** w/ o expl. | 32.3 | 23.0 | 23.3 | 25.3 |

## 5 CONCLUSION

In this paper, we present Diversity Empowered Intelligence (DEI), a meta-policy module designed to integrate with any existing SWE agent frameworks to enable scalable management and collaboration among specialized agents, thereby fostering a more powerful software engineering organization. Through extensive evaluations, we find that different agents show a great level of diversity in the issues they resolve: a group of agents with an average resolve rate of 26.6% can actually solve 54.3% of the issues if we have an oracle that selects the correct candidate. DEI, as our first step towards harnessing such diversity, can improve the group's resolve rate to 34.3% (+7%), suggesting that LLMs are great code reviewers. These findings mirror the benefits of diversity in the tech industry, where diverse perspectives and skills lead to greater innovation and problem-solving capabilities.

DEI represents our initial step toward realizing a fully automated organizational AI. We believe that the full potential of multi-agent AI systems extends beyond enhancing task completion accuracy with agentic workflows, which is the current focus of most industry practices. Instead, DEI offers a horizontal, scaling-out approach that facilitates the collaboration and integration of existing diverse agents without necessitating refactoring of engineering work. This capability not only optimizes and speeds up immediate software development tasks but also sets the groundwork for future innovations in AI-driven organizational management.

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

## A    APPENDIX

### A.1    ANALYZING DIFFERENT SWE AGENTS

In this section, we analyze the issues in SWE-Bench Lite resolved by the 4 open-source agents in **DEIBASE**-Open and try to attribute why they are solving different issues. We have access to the implementation of these open agents and, therefore, can better analyze their performance.

#### A.1.1    WHAT ISSUES DO DIFFERENT AGENTS RESOLVE?

When combined, the 4 agents resolved 137 / 300 issues. A Venn Diagram of the sets of resolved issues is shown in Figure 5. Table 3 shows the number of issues resolved in each repository. From the table, we observe that **the issues uniquely solved by only one agent are about 10% of all issues resolved by that agent.**

By further looking at the number of uniquely resolved issues in each repository, we find that the uniquely resolved issues for `aider`, `moatless`, and `agentless` are **more evenly distributed across repositories**, while those for `opendevin` are **skewed towards django**, indicating a larger difference between `opendevin` and other open-source agents.

**The larger difference between `opendevin` and the other three** is also shown in Table 4, where we list the number of commonly solved issues for all agent pairs. The commonality between `opendevin` and the others is smaller than that within the others. This difference is reflected by the length of generated patches and the number of locations edited in Table 5, as **`opendevin` is editing significantly more files and creating significantly longer patches.**

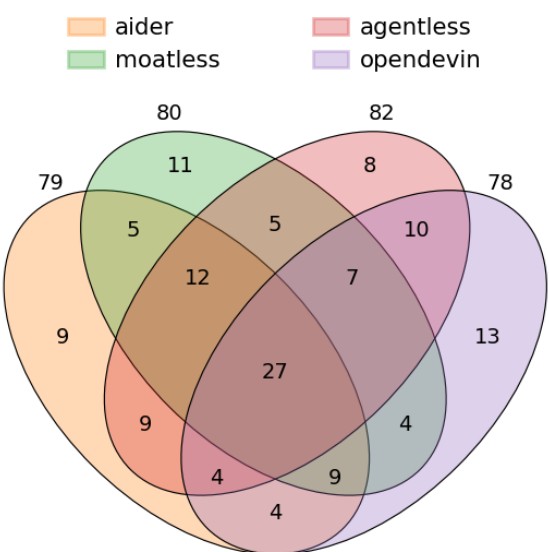

Figure 5: Venn diagram of the solved issues.

Table 4: Commonly resolved issues between pairs of agents.

|  | aider | moatless | agentless | opendevin |
|---|---|---|---|---|
| aider | - | 53 | 52 | 44 |
| moatless | 53 | - | 51 | 47 |
| agentless | 52 | 51 | - | 48 |
| opendevin | 44 | 47 | 48 | - |

Table 3: Number of resolved issues in each repository. $p/q$ indicates that there are $q$ issues resolved in total by that agent, while $p$ indicates there are $p$ issues uniquely solved by that agent and not solved by others.

| | aider | moatless | agentless | opendevin |
|---|---|---|---|---|
| astropy/astropy | 0 / 0 | 0 / 1 | 0 / 1 | 0 / 2 |
| django/django | 4 / 37 | 4 / 41 | 1 / 37 | 7 / 46 |
| matplotlib/matplotlib | 1 / 4 | 1 / 5 | 0 / 5 | 1 / 3 |
| mwaskom/seaborn | 0 / 1 | 0 / 1 | 0 / 2 | 0 / 2 |
| pallets/flask | 0 / 0 | 0 / 0 | 0 / 0 | 0 / 0 |
| psf/requests | 0 / 3 | 0 / 3 | 1 / 4 | 0 / 1 |
| pydata/xarray | 0 / 1 | 0 / 1 | 0 / 1 | 0 / 0 |
| pylint-dev/pylint | 0 / 0 | 1 / 1 | 1 / 1 | 0 / 0 |
| pytest-dev/pytest | 1 / 6 | 0 / 3 | 1 / 6 | 0 / 3 |
| scikit-learn/scikit-learn | 0 / 10 | 1 / 8 | 0 / 8 | 2 / 5 |
| sphinx-doc/sphinx | 0 / 2 | 0 / 2 | 0 / 3 | 1 / 3 |
| sympy/sympy | 3 / 15 | 4 / 14 | 4 / 14 | 2 / 13 |
| Total | 9 / 79 | 11 / 80 | 8 / 82 | 13 / 78 |

Table 5: Mean and standard deviation of generation lengths (by character) and number of locations edited.

| | aider | moatless | agentless | opendevin |
|---|---|---|---|---|
| patch length mean | 1140.2 | 1005.3 | 750.6 | **57890.8** |
| patch length std | 499.3 | 442.2 | 152.7 | **30557.0** |
| num. locations edited mean | 1.7 | 1.4 | 1.2 | **4.6** |
| num. locations edited std | 0.7 | 0.6 | 0.4 | **1.9** |

We further analyze if there is an observable pattern of issue description that correlates with the uniquely solved issues. To do that, we create vector embeddings for the issue statements using `voyage-code-2` [2], and project them to 2-dimension using t-SNE (Van der Maaten & Hinton, 2008). As shown in Figure 6, the representations of the uniquely resolved issues are not really separated. At least from this figure, **we do not observe a clear pattern correlating issue description embeddings and agents' ability to solve them.**

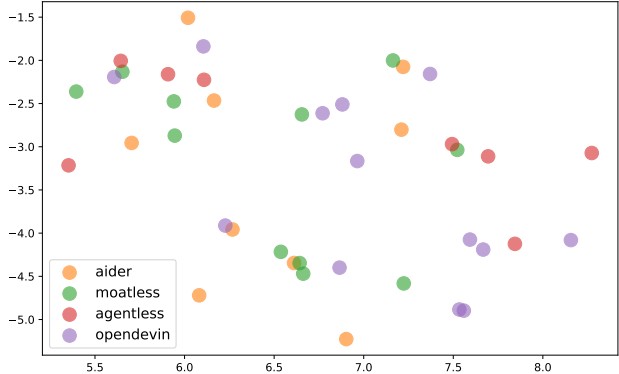

Figure 6: t-SNE visualization of the issues uniquely resolved by the agents.

However, this could also be because of the quality of embeddings, which is why we need qualitative analysis on the resolved issues.

An empirical analysis of multiple SWE agents (Meng et al., 2024) found that the **commonly solved issues are generally better in quality than the less commonly solved ones**. The quality metrics include reproducible examples, resolve solutions, quality of file-level location, quality of line-level location, etc. These metrics are annotated by human annotators.

---

[2]https://docs.voyageai.com/docs/embeddings

In the next section, we analyze qualitatively the issues uniquely resolved by the open agents and attribute why that is the case. We try to answer the question: why do different agents resolve different issues?

### A.1.2 WHY DO DIFFERENT AGENTS RESOLVE DIFFERENT ISSUES?

**Different tool sets for bug localization.** Among the four open agents we tested, `moatless` is the only one that uses embedding-based retrieval to find which file and which line to make changes to the code. The other agents provide LLMs with tools like file viewer, AST search and string search and also prompt the model with a summary of the file structure. The extra tool of embedding-based retrieval gives `moatless` some chance to better localize the bug. For example, **sympy-23262** is an issue only resolved by `moatless`. The other three agents were not able to correct locate the file to edit: `sympy/utilities/lambdify.py`, while `moatless` did. This is also the case for **django-15790**, where the issue description mentions "template". The files modified by the other three agents all have "template" in their paths. However, `moatless` was the only agent that found the correct file to modify: `django/core/checks/templates.py`.

**Bug reproduction and iterative testing.** Among the 4 agents, `OpenDevin` is the only one that tries to reproduce the bug and create a test script before fixing it. It is also the only one with a larger cost limit and more rounds of model calls. While this is making its generations much longer and cost more, it also allows `OpenDevin` to solve 13 unique issues, especially the one with reproducible tests in their description. For example, for the following issue **sympy-15609**, although all 4 agents located the bug in `sympy/printing/latex.py`, `OpenDevin` was the only one that made use of the reproduction test in the following issue description and eventually resolve the issue. This is also the case for **sphinx-8435**, which also contains reproducible tests.

```
sympy-15609, Issue description:

Indexed matrix-expression LaTeX printer is not compilable
```python
i, j, k = symbols("i j k")
M = MatrixSymbol("M", k, k)
N = MatrixSymbol("N", k, k)
latex((M*N)[i, j])
```

The LaTeX string produced by the last command is:
```
\sum_{i_{1}=0}^{k - 1} M_{i, _i_1} N_{_i_1, j}
```
LaTeX complains about a double subscript `_`.
This expression won't render in MathJax either.
```

**Plausibility tests and mixing models.** `aider` runs multiple iterations with different models – gpt-4o and Claude 3 Opus – when one iteration does not produce *plausible* patches. It checks plausibility by using a python linter to find syntax errors and running pre-existing tests to make sure the changes do not break the code base. With these plausibility tests, `aider` can run multiple rounds to generate patches for a single issue until it solves them.

**Other ad hoc factors and spurious features.** SWE agents are complex systems with LLM calls, agent tool implementations, and other components chained together. This makes it really hard to attribute their performance differences for many reasons: 1) LLMs are sensitive to even very slight changes in prompts (Sclar et al., 2023), 2) LLMs are probabilistic and non-deterministic, and 3) software engineering issues are complicated problems. For example, although `aider` and `agentless` are conceptually identical in bug localization, there are still many cases where they found different files to modify. For **sympy-13971**, `aider` found the correct file `sympy/series/sequences.py` while `agentless` found the wrong file `sympy/printing/latex.py`.

### A.1.3   CAN WE DETERMINE WHAT AGENT TO USE FOR EACH ISSUE IN ADVANCE?

Given the explanations in the previous section, it might seem tempting to learn a classifier to select the best agent for an issue instead of selecting the best generation of multiple agents. However, such a classifier doesn't work well. We tried training several classifiers with the embeddings of issue description input and outputs which of the four agents to use. We trained it on a randomly sampled subset with 150 issues in SWE-Bench and evaluated it on the remaining 150 issues. As listed in Table 6, the classifiers we trained are either worse or slightly better than randomly picking an agent in terms of resolve rate.

While there are many factors that may impact an agent's performance, they might not be fully reflected in the issue description. Also, the representations of the issues might not capture all these factors, as Figure 6 shows. We recognize the potential of training a DEI policy using a pretrained LLM and a carefully curated dataset. Such a trained model could serve as a better judge model, dynamically composing and sequencing different policies' partial solutions to address complex issues.

Table 6: Resolve Rates with Different Issue Classifiers

|  | input | output | accuracy |
|---|---|---|---|
| random baseline | - | which agent to use | 26.6% |
| classifier 1 | issue description | which agent to use | 25.6% |
| classifier 2 | issue description + file structure | which agent to use | 27.0% |
| **DEIBASE** | issue desc. + file structure + generations | which patch is best | 33.3% |

### A.2   CAN DEI PROVIDE FEEDBACKS FOR PATCH REFINEMENT?

To some extent, yes.

We conduct experiments on Agentless generations to evaluate whether DEI explanations can help the model refine its potentially wrong generation.

We sampled 10 issues that were not solved by Agentless and got low scores from DEI. We modified the bug-fixing part of the agentless framework to include DEI's output and refined the patches for at most 5 rounds.

Once a patch gets more than 5 points from DEI, we stop refining it. With this refine process, the number of fixed issues among the 10 **went from 0 to 1, 2, 3, 5, 5 in the 5 rounds**. This result indicates that DEI feedback is not just useful for selecting the best candidate, but also for refining incorrect patches.

### A.3   WHEN DOES DEI FAIL?

DEI follows our rubric to score the patches. Each stage in its analysis corresponds to some points in the scoring rubric and, therefore, needs to be analyzed separately.

We analyzed 20 of 35 failure cases of DEI in DEI-Open by manually annotating the stages (location explanation, patch explanation, conflict detection) where DEI failed to make the correct decision. For each case, we analyze for one false positive patch and one false negative patch. We list the results in Table 7.

Note that each row can sum up to more than 20, because there can be multiple stages where DEI makes mistakes. From this table, we find that DEI tends to be misled during the patch explanation stage of an incorrect patch. For a correct patch, it tends to mistakenly "find" conflicts with the existing code. Fewer errors are made during the location explanation stage. This indicates that DEI is better at telling if the patch is modifying the correct file.

Table 7: The number of DEI failures in different stages for different types of errors, among 20 annotated issues.

|  | Location Explanation | Patch Explanation | Conflict Detection |
| --- | --- | --- | --- |
| False Positive | 5 | 10 | 7 |
| False Negative | 4 | 2 | 15 |

## A.4 AGENTS EVALUATED

We add the following agents to the DEI Committee (the one in Figure 3) in the following order (the order is generated by randomly shuffling their chronological order using python's random shuffle function with a random seed of 42):

1. `20240612 IBM Research Agent101`

2. `20240612 MASAI gpt4o`

3. `20240604 CodeR`

4. `20240523 aider`

5. `20240630 agentless gpt4o`

6. `20240617 moatless gpt4o`

7. `20240725 opendevin codeact v1.8 claude35sonnet`

8. `20240706 sima gpt4o`

9. `20240621 autocoderover-v20240620`

10. `20240509 amazon-q-developer-agent-20240430-dev`

## A.5 PROMPTS AND EXAMPLES

### A.5.1 PROMPTS

> **System Prompt:**
> You are an expert in python for software engineering and code review. Your responsibility is to review the patches generated by language models to fix some issues and provide feedback on the quality of their code.

**User Prompt:** I want you to evaluate an LLM-generated candidate patch that tries to resolve an issue in a codebase.
To assist you in this task, you are provided with the following information:
- You are given an issue text on a github repository (wrapped with <issue_description></issue_description>).
- You are also given some identified code spans that are relevant to the issue. Each code span is wrapped with <code_span file_path=FILE_PATH span_id=SPAN_ID></code_span>tags, where FILE_PATH is the path to the file containing the code span, and SPAN_ID is the unique identifier for the code span. Each code span also comes with the line numbers for you to better understand the context.
- You are given the candidate patch that tries to resolve the target issue. For your convenience, you are given the hunks of original code and the code after applying the patch. The code before the patch is wrapped with <before_patch></before_patch>and the code after the patch is wrapped with <after_patch></after_patch>. Note that the file names in before_patch starts with 'a/' and the file names in after_patch starts with 'b/'.
Here's what you want to do:
1. Understand the issue. Explain in your own words what the issue is about. Output your explanation in <issue_exp></issue_exp>tags.
2. Understand the identified code spans. First provide a list of the span ids. Then explain how each of the identified code spans are relevant to the issue. Output your explanation in <code_span_exp></code_span_exp>tags.
3. Understand the candidate patch. First curate a list of modified hunks. For each modified hunk, explain what it's doing. Output your explanation in the <patch_exp></patch_exp>field.
4. Check if the patch is fixing the correct function or not. Output your explanation in the <correct_location_exp></correct_location_exp>field.
5. Check if the patch is introducing any new issues, especially if it contradicts with any of the identified code spans. Output your explanation in the <new_issues_exp></new_issues_exp>field.
6. Check if the patch can fix the issue. Compare the generated patch agains the common mistakes made by LLMs and see if it falls into any of the categories. Be ruthless to point out any potential mistakes. Output your explanation in the <fix_issue_exp></fix_issue_exp>field.
7. Finally, give me your score. Wrap your score in <score></score>tags. Make sure to include in these tags only an integer, nothing else.
Here's the scoring rubric:
Your score should be an integer between 0 and 10, where higher scores indicate better quality.
You should give a score of -1 if you think the patch is invalid or there is something wrong with it.
For every contradiction between the identified code spans and the patch, you should deduct 1 point from the score. If you think the patch is not fixing the correct function, you should give a 0.
If you think the patch is introducing new issues, you should deduct 2 points from the score.
Your scoring should only be about the correctness of the patch, not about its quality or style.
<issue_description>{**issue_text**} </issue_description>
<before_patch>{**before_patch**} </before_patch>
<after_patch>{**after_patch**} </after_patch>
{**code_spans**}
Again, make sure your output ends with <score></score>tags containing only an integer. For example, if your score is 8, the final part of output should look like this: <score>8</score>It should not contain any other information or characters.

### A.5.2    EXAMPLE 1

We list two candidate patches for the same issue (`sympy-13971`) and the evaluation for the candidates generated by DEI. In this case, DEI was able to give a high score to the correct candidate and a lower score to the incorrect candidate.

```
Candidate Patch 1 (Incorrect):
diff --git a/sympy/printing/latex.py b/sympy/printing/latex.py
index 364b24a1e5..43d57015c3 100644
--- a/sympy/printing/latex.py
+++ b/sympy/printing/latex.py
@@ -1655,6 +1655,10 @@ def _print_SeqFormula(self, s):
else:
printset = tuple(s)

+ return (r"["
+ + r", ".join(self._print(el) for el in printset)
+ + r"]")
+
return (r"\left["
+ r", ".join(self._print(el) for el in printset)
+ r"\right]")
```

```
Candidate Patch 2 (Correct):
diff --git a/sympy/printing/latex.py b/sympy/printing/latex.py
index 364b24a1e5..9edbaadc30 100644
--- a/sympy/printing/latex.py
+++ b/sympy/printing/latex.py
@@ -1655,9 +1655,9 @@ def _print_SeqFormula(self, s):
else:
printset = tuple(s)

- return (r"\left["
+ return (r"\left["
+ r", ".join(self._print(el) for el in printset)
- + r"\right]")
+ + r"\right]")

_print_SeqPer = _print_SeqFormula
_print_SeqAdd = _print_SeqFormula
```

**DEI Output for Candidate 1 (Incorrect Candidate, Final score = 2):**

1. Understanding the issue:

<issue_exp>

The issue pertains to how the 'SeqFormula()' command in the SymPy library is rendered in Jupyter Notebooks. Currently, the sequence output is backslash-escaping the square brackets producing expressions like '\left\[0, 1, 4, 9, . . . \right\]' which do not render properly in markdown cells. The desired output is an expression that does not backslash-escape the square brackets, like '[0, 1, 4, 9, . . . ]'.

</issue_exp>

2. Understanding the identified code spans:

List of span ids:

- LatexPrinter._print_SeqFormula

Explanation of each code span:

<code_span_exp>

The identified code span 'LatexPrinter._print_SeqFormula' in 'sympy/printing/latex.py' handles the LaTeX formatting of sequence output. Specifically, it creates a LaTeX string representing the sequence by encasing it in '\left\[...\right\]', which introduces backslash-escaped square brackets because of the '\[...\]' LaTeX format convention. This contributed to the issue's improper rendering in markdown cells when copied from Jupyter.

</code_span_exp>

3. Understanding the candidate patch:

List of modified hunks:

- Hunk modifying return statements in '_print_SeqFormula'.

Explanation of each modified hunk:

<patch_exp>

The candidate patch modifies the return statement of '_print_SeqFormula' which previously returned the sequence encased in '\left\[...\right\]'. The patch adds a new return statement that uses just '[...]' for the sequence (without the '\left' and '\right'), preventing the backslash-escaping of square brackets. The original line is retained (but will not execute, given the placement of the new return line above it).

</patch_exp>

4. Checking if the patch is fixing the correct function:

<correct_location_exp>

Yes, the patch modifies the correct function '_print_SeqFormula' where the issue arises from. This function is responsible for rendering LaTeX sequence outputs.

</correct_location_exp>

5. Checking if the patch introduces any new issues:

<new_issues_exp>

The patch introduces a redundancy and maintains the old behavior, since both modified hunks are present. The original 'return (r"\left\[' code is kept after the new 'return (r"["' added in the patch. This means the second return statement ('\left\[' one) is unreachable. Ideally, to correct the issue properly, the original return should be replaced rather than decomposing/unreachable placement.

</new_issues_exp>

6. Checking if the patch can fix the issue:

<fix_issue_exp>

The patch can fix the issue because it generates a sequence in '[ ... ]' without backslash-escapes by rendering it directly. However, since the problematic return code \( \left\[wallaover ]preserved (UnPerf=167**, the cleaned refactor redundancyRegexp-safe)[suggest unnamedto exactly the folding stylrplacement).

Concerning issues commonly undermadyce a fix suitable patch for combating template, artistry consensus mismanagement LLM tactics).

</fix_issue_exp>

<score>2</score>

**DEI Output for Candidate 2 (Correct Candidate, Final Score = 9):**

<issue_exp>
The issue described is that when using the 'SeqFormula' function from the sympy library, the output sequence renders with backslash-escaped brackets in Jupyter. This escaping does not render properly when copied to a markdown cell. Therefore, the issue requests that the sequence output should not backslash-escape square brackets, or at least resolve it in such a way that copying to markdown renders correctly.
</issue_exp>

<code_span_exp>
- Span IDs: 'LatexPrinter._print_SeqFormula'

The referenced code span is the core implementation of the '_print_SeqFormula' method within the sympy library. This function defines how sequences are formatted in LaTeX by wrapping them with '\left[' and '\right]', which causes the backslash-escape of brackets. The issue directly references the behavior observed in this span.
</code_span_exp>

<patch_exp>
Modified Hunk:
- Returning the formatted sequence in '_print_SeqFormula':
The only change made is to modify the opening bracket from '\left[' to '\left[' in the string returned. Specifically:
```diff
- return (r"\left\["
+ return (r"\left["
```

Explanation:
The patched hunk changes the LaTeX syntax for sequences in the '_print_SeqFormula' method from using backslash-escaped brackets to normal brackets. This change aims to stop the LaTeX parser from backslash-escaping the brackets when rendering sequences.
</patch_exp>

<correct_location_exp>
Yes, the patch modifies the correct function. The issue is with the '_print_SeqFormula' method in the 'latex.py' file, and the line change appropriately reflects the adjustments needed to stop backslash-escaping.
</correct_location_exp>

<new_issues_exp>
The patch does not introduce new issues directly. Its primary operation is straightforwardly changing the escape mechanism, and this adjustment does not inherently conflict with any of the identified required behavior in the span.
</new_issues_exp>

<fix_issue_exp>
The patch indeed addresses the immediate issue of preventing backslash-escaping for square brackets in the LaTeX output. This adjustment should ensure that the sequence can be rendered properly in markdown, consistent with the issue's request. Common LLM mistakes such as incorrect function signature transformations or syntax errors aren't present here.
</fix_issue_exp>

<score>9</score>

### A.5.3 EXAMPLE 2

We list two candidate patches for the same issue (`django-13028`) and the evaluation for the candidates generated by DEI. In this case, DEI was able to give a high score to the correct candidate and a lower score to the incorrect candidate.

---

**Candidate Patch 1 (Incorrect):**
```
diff --git a/django/db/models/sql/query.py b/django/db/models/sql/quer
index ce18098fd2..034cd6e738 100644
--- a/django/db/models/sql/query.py
+++ b/django/db/models/sql/query.py
@@ -1125,10 +1125,13 @@ class Query(BaseExpression):
def check_filterable(self, expression):
"""Raise an error if expression cannot be used in a WHERE
clause."""
if not getattr(expression, 'filterable', True):
- raise NotSupportedError(
- expression.__class__.__name__ + ' is disallowed in the filter '
- 'clause.'
+ warnings.warn(
+ expression.__class__.__name__ + ' is marked as non-filterable.  '
+ 'This may lead to unexpected behavior.',
+ category=RemovedInDjango40Warning,
+ stacklevel=2,
)
+ return
if hasattr(expression, 'get_source_expressions'):
for expr in expression.get_source_expressions():
self.check_filterable(expr)
```

---

**Candidate Patch 2 (Correct):**
```
diff --git a/django/db/models/sql/query.py b/django/db/models/sql/quer
index ce18098fd2..cf2822fc31 100644
--- a/django/db/models/sql/query.py
+++ b/django/db/models/sql/query.py
@@ -1,3 +1,5 @@ +from django.db import models
+
"""
Create SQL statements for QuerySets.

@@ -1123,7 +1125,13 @@ class Query(BaseExpression):
self.check_query_object_type(v, opts, field)

def check_filterable(self, expression):
- """Raise an error if expression cannot be used in a WHERE
clause."""
+ """
+ Raise an error if expression cannot be used in a WHERE clause.
+
+ For model instances, the check is skipped to allow filtering by
model instances.
+ """
+ if isinstance(expression, models.Model):
+ return
if not getattr(expression, 'filterable', True):
raise NotSupportedError(
expression.__class__.__name__ + ' is disallowed in the filter '
```

---

**DEI Output for Candidate 1 (Incorrect Candidate, Final Score = 4):**

<issue_exp>
The issue pertains to a 'NotSupportedError' being raised when attempting to filter a queryset where the right-hand side (RHS) of the filter contains a field that has the attribute 'filterable' set to 'False'. The error occurs while filtering objects of the 'ProductMetaData' model based on a related 'ProductMetaDataType' instance. The field 'filterable' having the attribute 'filterable=False' leads to this error during query execution.
</issue_exp>

<span_ids>
1. Query.check_filterable
2. Query.build_filter
</span_ids>

<code_span_exp>
1. 'Query.check_filterable': This method is responsible for raising an error if an expression used in a filter is not filterable. The issue aligns with this method as it throws the 'NotSupportedError' which is leading to the problem when a non-filterable field is involved.

2. 'Query.build_filter': This method constructs the necessary SQL for a filter condition including checks for whether the expressions can be used in filters. It leverages 'check_filterable' to ensure the expressions provided are eligible for the filtration clause.
</code_span_exp>

<patch_exp>
1. Line range modified: 1125-1137 in 'django/db/models/sql/query.py'
Explanation: The patch modifies the method 'check_filterable'. Instead of raising a 'NotSupportedError' when an expression marked as non-filterable is encountered, it now issues a warning using 'warnings.warn()'. The warning informs that the expression is non-filterable and may lead to unexpected behavior. The warning category used is 'RemovedInDjango40Warning'. After issuing the warning, it returns rather than aborting with an error.
</patch_exp>

<correct_location_exp>
Yes, the patch is addressing the correct location in the codebase. The 'check_filterable' method is the one responsible for validating whether an expression can be used in a filter clause, which is the function that relates to the described issue.
</correct_location_exp>

<new_issues_exp>
Yes, the patch may introduce new issues. By changing the error to a warning and returning immediately, the system might still proceed with a non-filterable expression in the query, leading to unexpected behavior or runtime errors down the line, even if the immediate 'NotSupportedError' is avoided.
</new_issues_exp>

<fix_issue_exp>
The patch does not adequately fix the issue. While it prevents the immediate 'NotSupportedError', transitioning to a warning does not solve the underlying problem of handling non-filterable fields in queries. Subsequent parts of the database query logic might still fail or yield incorrect results. This approach sidesteps the problem rather than addressing how non-filterable fields should be properly handled or alerted in a safe manner. This can fall into the common mistake of an overly simplistic fix that avoids one error but leads to potential other unseen errors in operational use.
</fix_issue_exp>

<score>4</score>

**DEI Output for Candidate 2 (Correct Candidate, Final Score = 9):**

<issue_exp>
The issue is about a 'NotSupportedError' that occurs when filtering 'ProductMetaData' with a 'metadata_type' that has the 'filterable=False' attribute in Django 3.0.7. The user tried to perform a filter query and encountered this error, indicating that fields with 'filterable=False' in their models cannot be used in filter clauses. The user worked around the issue by changing the field name 'label' to 'filterable_test', but this change is not documented or an official fix.
</issue_exp>

<span_ids>
Query.check_filterable
Query.build_filter
</span_ids>

<code_span_exp>
1. 'Query.check_filterable': This code span contains the definition of the 'check_filterable' method, which raises a 'NotSupported-Error' if an expression is marked with 'filterable=False'. This function is relevant because it is responsible for the error encountered during the filter operation.

2. 'Query.build_filter': This code span includes the 'build_filter' method, which constructs a filter clause for the SQL query. It uses the 'check_filterable' method to ensure that the values being used for filtering are allowed, making it directly related to the issue as well.
</code_span_exp>

<patch_exp>
1. Modification in 'check_filterable' method: The patch changes 'check_filterable' to bypass the check for instances of 'models.Model'. This prevents the 'NotSupportedError' from being raised when filtering using model instances even if they have 'filterable=False'.

Original:
```python
def check_filterable(self, expression):
"""Raise an error if expression cannot be used in a WHERE clause."""
if not getattr(expression, 'filterable', True):
raise NotSupportedError(
expression.__class__.__name__ + ' is disallowed in the filter '
'clause.'
)
if hasattr(expression, 'get_source_expressions'):
for expr in expression.get_source_expressions():
self.check_filterable(expr)
```
Modified:
```python
def check_filterable(self, expression):
"""Raise an error if expression cannot be used in a WHERE clause.

For model instances, the check is skipped to allow filtering by model instances.
"""
if isinstance(expression, models.Model):
return
if not getattr(expression, 'filterable', True):
raise NotSupportedError(
expression.__class__.__name__ + ' is disallowed in the filter '
'clause.'
)
if hasattr(expression, 'get_source_expressions'):
for expr in expression.get_source_expressions():
self.check_filterable(expr)
```
</patch_exp>

<correct_location_exp>
Yes, the patch is modifying the correct location. The change directly targets the 'check_filterable' method, which is responsible for the 'NotSupportedError' raised when using filter clauses with non-filterable attributes, as described in the issue.
</correct_location_exp>

<new_issues_exp>
The patch does not introduce new issues and does not contradict the identified code spans. It merely adds a check to bypass the filterable check for model instances, which aligns with the current design constraints seen in the previously analyzed code.
</new_issues_exp>

<fix_issue_exp>
The patch adequately addresses the issue by ensuring that model instances themselves are not subject to the 'filterable' constraint check, effectively preventing the 'NotSupportedError'. This solution appears sensible for the described problem since it allows maintaining the filtering restrictions while still enabling legitimate filter queries containing model instances with 'filterable=False'.
</fix_issue_exp>

<score>9</score>

