# OpenReview forum: "Diversity Empowers Intelligence: Integrating Expertise of Software Engineering Agents"
_ICLR.cc/2025/Conference — ICLR 2025 Poster_

### Official Review · Reviewer_zt1v · 2024-10-25

**Soundness:** 3
**Presentation:** 3
**Contribution:** 3
**Rating:** 8
**Confidence:** 4

**Summary:**

The authors propose “Diversity Empowered Intelligence,” a method for aggregating the solutions of several SWE agents by means of LLM prompting. They show a significant increase in scores on SWE-Bench Lite and attribute this to the fact that different SWE agents can solve different subsets of the problems.

**Strengths:**

[1] The authors systematically explore the benefits of ensembling of SWE agents and show significant improvement over single-agent solutions.

[2] Figure 1 nicely illustrates the idea behind DEI. Table 1 is very insightful and provides strong evidence of the value of DEI.

**Weaknesses:**

[1] The results are presented for SWE-Bench only. Even though SWE-Bench is a fairly reliable and diverse benchmark, validating DEI on another SWE benchmark could solidify the claim.

[2] The improvement in scores comes at the cost of significantly higher token usage (or compute).

[3] The following papers are closely related but not cited:

a) SELF-CONSISTENCY IMPROVES CHAIN OF THOUGHT REASONING IN LANGUAGE MODELS

https://arxiv.org/pdf/2203.11171

Since a common decision is made based on a diverse set of candidates.

b) Judging LLM-as-a-Judge with MT-Bench and Chatbot Arena

https://arxiv.org/abs/2306.05685

Since the scoring by an LLM is prompted, which is essentially LLM-as-a-Judge.

[4] The metrics in 4.1.2 are tied to the order in which the agents (and, by extension, their proposed solution patches) are added to the list. This is unacceptable for a metric as the solutions are fundamentally an unordered set. The metrics should be refined to be permutation invariant towards the set of agents. Whereas the proposed metrics are flawed in the general case, the main results in section 4.2 that are collected with the specific ordered list of agents are not flawed and have analytical value.

**Questions:**

[1] In “4.1.2 Evaluation Mertics” for Union@k: “Solved by any of the k solutions.”, what is the mathematical definition of “any”?

[2] Likewise, for Intersect@k, “solved by all k solutions”: is k tied to the number of agents at hand, or is it a free parameter? If independent, which agents are chosen from the (unordered) set of agents?

---

> ### Comment · Reviewer_zt1v · 2024-11-26
>
> I encourage the authors to address my questions and outlined weaknesses to ensure that I keep my score.

---

> > ### Author Response · Authors · 2024-11-26
> > **Rebuttal Response Available in 1 Day**
> >
> > Dear Reviewer zt1v,
> >
> > We greatly appreciate your thorough and constructive comments, many of which have helped improve our paper.
> >
> > Particularly, the comment about the **permutation invariance of evaluation metrics** has enlightened us to add newly conducted experiments to properly address it. We've been working on the new experiments since the beginning of rebuttal and will have the response for your review very soon (hopefully, within 1 day). Thanks again for your encouraging feedback on the original submission! We are very grateful that you could give us additional time for response.
> >
> > With best regards,
> >
> > Authors of submission 9222

---

> ### Author Response · Authors · 2024-11-27
>
> Dear Reviewer zt1v,
>
> Thank you for your patience. Here's our response. We've also revised our manuscript (in blue text) to address your suggestions.
>
> ### W1: validating DEI on another SWE benchmark
>
> We understand your concern about more diverse evaluations.
>
> However, we were not able to do that because realistic agentic benchmarks in software engineering are scarce and only SWE benchmark has the multiple agents we need to use as candidates for DEI.
>
> Although it is hard for us to do the same experiments because of the time limit and lack of benchmark,, we do plan to extend DEI to other agentic-tasks that have diverse solutions.
>
> ### W2: the cost of significantly higher token usage
>
> To address your concern, we provide a cost analysis for DEI. There are two parts of extra cost brought by DEI -- running more agents and evaluating multiple candidates.
>
> If we consider the open agents in DEI-Open, running an agent for a single issue costs about **\$0.5**. Evaluating a generated patch with 10 votes costs about **\$0.1**. Therefore, adding one more candidate costs **\$0.6** in total. We argue that **such cost is tolerable for resolving an issue**, considering the amount of money it takes for human programmers to resolve an issue.
>
> Furthermore, **we suggest a potential mitigation to the cost problem**: although we have the same number of agents in the committee for each issue, this can be further optimized. We can save the extra cost for more candidates by only adding candidates when DEI finds the initial generation not satisfactory.
>
> ### W3: Related papers not cited
>
> Thank you for pointing them out. These are indeed related to our method.
>
> We have added citations to these papers you suggested in our revised manuscript.
>
> ### W4: Metrics are tied to the order of agents
>
> Using permutation-invariant metrics is a great suggestion. We've changed our definition of all the @k metrics in the revised **section 4.1.2**.
>
> Specifically, instead of calculating these metrics for the first k agents according to a fixed order, we calculate their expectation and standard deviation for the first k agents over all permutations.
>
> This calculation is equivalent to calculating the metric for all possible k-element subsets of the candidate set with n elements, which can be done in $O(2^n)$ time.
>
> We've updated our main result **Figure 3** according the modified definition. We use concrete dots and lines to represent expectations and shaded regions to represent standard deviations. With the modified definition, the improvements brought by DEI are clearer and better justified.
>
> For example, the new results for the 10 different agents are reported in the following table:
>
> | k    | 1    | 2    | 3    | 4    | 5    | 6    | 7    | 8    | 9    | 10   |
> | ---- | ---- | ---- | ---- | ---- | ---- | ---- | ---- | ---- | ---- | ---- |
> | Union@k | 26.6 | 36.1 | 41.3 | 44.8 | 47.4 | 49.3 | 50.9 | 52.2 | 53.3 | 54.3 |
> | Intersect@k | 26.6 | 17.1 | 12.9 | 10.5 | 8.8  | 7.6  | 6.7  | 5.9  | 5.2  | 4.7  |
> | Average@k | 26.6 | 26.6 | 26.6 | 26.6 | 26.6 | 26.6 | 26.6 | 26.6 | 26.6 | 26.6 |
> | DEI@k | 26.6 | 30.2 | 31.7 | 32.7 | 33.4 | 34.0 | 34.4 | 34.9 | 35.3 | 35.7 |
>
> We are really glad you suggested this.
>
> ### Q1 and Q2: solved by any/all of k solutions?
>
> **k is the number of candidates that is a free parameter independent from the total number of candidates at hand.**
>
> As we mentioned in our response to W4, we consider all sets of size k that are subsets of a larger n-element set of candidates.
>
> > which agents are chosen from the (unordered) set of agents
>
> After modifying our metrics to be permutation-invariant, we consider all possibilities instead of a fixed order of agents. Thank you for raising these questions so that we can make our manuscript clearer.
>
> We thank you for acknowledging our work and providing important suggestions for improvement. We believe your feedback has greatly improved our paper. Please do not hesitate to ask if you have any further concerns. We would love to address them.

---

### Official Review · Reviewer_aexB · 2024-10-30

**Soundness:** 2
**Presentation:** 3
**Contribution:** 2
**Rating:** 5
**Confidence:** 4

**Summary:**

The paper focuses on software engineering (SWE) agents, i.e., AI agents, in support of various software engineering tasks, such as code generation, automated testing, and project management. Specifically, in this work, the authors focus on AI agents' ability to resolve GitHub issues based on their descriptions. Observing that different SWE agents resolve different sets of issues successfully despite having similar resolve rates, the authors propose a method for organizing various SWE agents into ensembles with the intent to use the union of their expertise in resolving GitHub issues. The proposed method, Diversity Empowered Intelligence (DEI), is evaluated using realistic issue tracker data. The results show superior performance compared to single SWE agents' performances.

**Strengths:**

- The topic is important. There is a clear emerging need for systematic methods for using LLMs, especially for software engineering.
- The method seems to be novel.
- The quality of the experiments is solid.

**Weaknesses:**

- The method is not exactly systematic, which limits its applicability and forces an ad-hoc way of working with ensembles of AI agents.

  - Sec 3.3.2, Equation 3: It is clear at this point that diversity is not purposefully engineered but rather: it's an emergent property of the system. One then has to wonder if this approach can be purposefully used in any problem or if its success will be rather non-deterministic and dependent on the problem at hand and its parameters (~ context).

  - Sec 3.3.2, L214-215: "The DEI framework leverages the strengths of each agent in their respective contexts to enhance overall performance across all contexts." - How does the framework account for intra-agent diversity? It seems that Figure 1 is rather deterministic, though some non-determinism is likely.

- Often vague formal underpinnings

  - Sec 3.3.1 is rather vague to provide formal underpinnings to the contributions. For example, L182: "The context space, C, includes relevant repository information and issue descriptions." What does "relevant information" entail here?

- Potentially limited mitigation of inconsistent agent behavior

  - Sec 3.2: "We consider two types of diversity: intra-agent diversity and inter-agent diversity." - I don't entirely agree with the proposed term of "intra-agent diversity". Diversity seems to be an artifact of non-determinism in this case, i.e., basically a temporal inconsistency. As such, it requires different means to mitigate, compared to the diversity stemming from the number of different agents. The method does not seem to acknowledge this.


- Typo: L057, citation "Team, 2024"

**Questions:**

- Sec 4.4: Are these results statistically significant?

- Is it possible to add human developers to the committee? (Fig 1c.)

---

> ### Author Response · Authors · 2024-11-25
>
> Dear Reviewer aexB,
>
> Thank you for the thoughtful comments.
>
> ### W1: diversity is not purposefully engineered but rather an emergent property. How does the framework account for intra-agent diversity?
>
> We argue that the diversity is purposefully engineered since we intentionally incorporate different policies, i.e. different SWE agent implementations for inter-agent diversity. They have different workflows, different tool sets, different prompts and different foundation models. It is expected that their generations will be different. What we try to do is inspect how large the difference is and exploit the difference.
>
> We appreciate your careful reading of the paper. Indeed, we need to better phrase our method section to also account for intra-agent diversity. Intra-agent diversity comes from the non-deterministic nature of language models, because the LLMs (gpt-4o, claude 3.5) in these frameworks are using non-deterministic sampling with a non-zero temperature to generate their outputs. This is also a deliberate choice because higher temperature tends to lead to more diverse outputs and better average performance with larger sample sizes [1].
>
> ### W2: What does "relevant information" entail here?
>
> As we have shown in the prompt example in the prompt examples in **Appendix A.5** and L265**,** there are 3 parts of relevant information:
>
> - Issue description, which is taken directly from SWE-Bench’s problem statement.
> - Relevant context, which is taken from the reasoning traces of the agents. It includes the files, and code snippets that are identified by the agent to be relevant to its generation.
> - Code before/after the patch. It includes the original code before the agent’s modification and afterwards, so that DEI can evaluate whether the changes are reasonable.
>
> ### W3: Intra-agent diversity is a temporal inconsistency that needs to be mitigated with different means.
>
> As we discussed in the response to W1, intra-agent diversity comes from LLMs’ non-deterministic nature. Such diversity (or temporal inconsistency) has often been mitigated using some type of reranker/evaluator in LLM research, such as self-consistency [2], best-of-n sampling [3, 4], minimum-bayes risk decoding [5], etc. In these previous studies, they are using rerankers to mitigate temporal inconsistency from LLMs. Our approach shares the same spirit as these methods.
>
> **Reference**
>
> [1] Evaluating Large Language Models Trained on Code, https://arxiv.org/abs/2107.03374
>
> [2] Self-Consistency Improves Chain of Thought Reasoning in Language Models, https://openreview.net/forum?id=1PL1NIMMrw
>
> [3] Lever: Learning to verify language-to-code generation with execution, https://proceedings.mlr.press/v202/ni23b/ni23b.pdf
>
> [4] Measuring goodharts law, https://openai.com/index/measuring-goodharts-law/
>
> [5] Improving the minimum Bayes’ risk combination of machine translation systems, https://aclanthology.org/2013.iwslt-papers.4/
>
> ### W4: Typo.
>
> We have fixed this in our revised manuscript.
>
> ### Q1: Are these results in Sec 4.4 statistically significant?
>
> If we understand correctly, you are referring to the ablations in Section 4.3.
>
> We ran the ablation experiments multiple times to measure their variance. Here’s an updated version of Table 2 with standard deviations:
>
> | | Open Agents | Agentless | Aider | Moatless |
> |---|---|---|---|---|
> | DEIBASE w/ expl. | 34.5 (std=0.8) | 26.0 (std=1.2) | 24.6 (std=0.7) | 25.6 (std=0.5) |
> | DEIBASE w/o expl. | 32.3 (std=0.6) | 23.0 (std=0.5) | 23.3 (std=0.6) | 25.3 (std=0.9) |
>
> Note that the std intervals have no overlaps for the first 3 groups, indicating that the results are statistically significant.
>
> ### Q2: Is it possible to add human developers to the committee?
>
> This is a great question. While it’s beyond the scope of this paper, it’s definitely possible to add human to the loop of DEI, because some mistakes in the generations are easily detectable by humans, especially contributors to these repositories. We imagine that replacing some parts of DEI with human programmers can improve its performance.
>
> We really appreciate your efforts in helping us refine our paper. Should you have any further questions, please do not hesitate to let us know.

---

> > ### Author Response · Authors · 2024-11-27
> > **Could you please let us know whether our responses and updated submission properly addressed your concern?**
> >
> > Dear Reviewer aexB,
> >
> > Thank you very much for your time spent on our submission and your questions. We have tried to address your concerns in the response and updated submission – any feedback from you would be appreciated. If you have further comments, please kindly let us know – we hope for the opportunity to respond to them.
> >
> > Best wishes, Authors of paper 9222

---

> > > ### Comment · Reviewer_aexB · 2024-11-28
> > >
> > > Thank you for your response; it indeed clarified things.

---

> > > > ### Author Response · Authors · 2024-11-28
> > > > **Thank You for Your Feedback and Re-evaluation Request**
> > > >
> > > > Dear Reviewer aexB,
> > > >
> > > > Thank you for your thoughtful feedback and response. We are pleased that our clarifications have addressed your concerns. If you find all issues resolved, we would appreciate your consideration in updating the evaluation score. We’re happy to provide any additional clarification needed.
> > > >
> > > > Thank you again for your valuable input.
> > > >
> > > > Best wishes, Authors of paper 9222

---

### Official Review · Reviewer_wcfY · 2024-11-03

**Soundness:** 3
**Presentation:** 3
**Contribution:** 3
**Rating:** 6
**Confidence:** 3

**Summary:**

The paper proposes using an esamble of agents to increase the number of software engineering (SWE) problems that can be solved showing that using different agents for different problems is better than using a single agent for all. The approach is based on a reinforcement learning formulation.

**Strengths:**

* Important problem with significant real world value.
* The paper is well written and easy to follow.
* The evaluation clearly shows both the added value of the approach and how much better it could be.
* Considers both intra and inter agent diversity.
* Good potential to extend the approach, see questions.

**Weaknesses:**

* It is unclear what new knowledge we have gained from the paper. That diversity is good is expected. To select the best candidate for each individual problem seems also expected. What is the unexpected or novel result of the paper? Or is the main result that you managed to get to work? - The authors clarified that it is how easy it is to achieve diversity and how significant it is.
* The significance of the approach might be on the low end. Since the approach is effective but straight forward, more elaborate analysis and/or experimentation would be expected. - Based on the authors response, the significance is higher than I initially estimated, but still somewhat limited.
* Some things are unclear, see questions. - Most of these were addressed by the authors.

**Questions:**

1) It seems to me that Eq. 3 expects the context to never change. Is that correct? (You fix the context, then you select the optimal policy for that context and then you use the expected reward from that policy.)
2) How do you select the optimal policy for a given context? Do you compute the expected reward for every policy given that reward and selected the highest? Or is this what is done in Step 3 Patch scoring?
3) Is there any learning involved in computing the DEI policy? It seems to me that the policies are given and then it is a deterministic computation to select the best of those.
4) Would it be possible to identify what the missing expertise is? I.e., if there was an agent capable of doing X, then the score would be significantly higher. There seems to be a potential to identify contexts that are far away from the contexts covered by the current policies.
5) Have you considered composed policies? Maybe a but is complex and would require two different patches, i.e. first apply partial solution by agent a_i and then apply partial solution by agent a_j.
6) If you have as many members on the committee as you have different agents, wouldn't you always get the oracle answer?
7) If the statement above is correct, wouldn't the interesting question be: What is the minimal number of members on the committee to get an acceptable score? What is the cost ($) benefit (resolve %) trade-off? The results would be much more interesting if the cost of running the different options would be included. It seems that you should be able to get a very clear analysis which would allow developers to estimate how much it would be worth to them to fix an issue.

---

> ### Author Response · Authors · 2024-11-25
> **Response Part 1/3**
>
> Dear Reviewer wcfY,
>
> We sincerely thank you for the time dedicated to reviewing our paper and the helpful comments. Here’s our response to your comments and questions.
>
> ### W1: What is the unexpected result of the paper? Or is the main result that you managed to get to work?
>
> As we mentioned in our contributions (L96-97) and experiment findings (L375), the unexpected result is not the generic idea of “diversity is good”, but **how surprisingly significantly diversity helps, and how easy it is to exploit such diversity through our orchestration layer**. We would like to clarify these findings here.
>
> First, despite the high level of similarity in SWE agents, they solve very different issues. In Figure 3a, 10 agents that solve around 26% - 30% of the issues, when combined, can solve ~54%, **almost 2x as any of the single agents**. Note that these agents are not that different from each other. **Their base models are similar**, either Claude Sonnet 3.5 or GPT-4o. **Their workflows are similar** and usually have bug localization and bug fixing as two stages. **The tools they provide the model with are similar.** They usually include string search, AST search, in-context repository overview, and embedding-based retrieval. **The ways they enable the model to edit the code are similar.** Most agents let the model edit by searching and replacing, rather than generating the actual diff format. It is surprising to me that despite all the features these agents share, their solved sets are so different. This finding alone tells us how not robust current LLMs can be, and how much potential there is to exploit the diversity.
>
> Second, as you have pointed out, our method is very straight-forward. We simply run multiple agents independently and combine their solutions through basic voting. We don't use any sophisticated ensemble techniques, model combinations, or complex aggregation methods. The fact that such a simple approach yields significant improvements (nearly 1.4x performance for the 10-agent group) highlights how easily exploitable this diversity is. This is particularly noteworthy because it suggests that even more sophisticated combination methods could potentially yield even better results.
>
> Finally, our findings have broader implications for the field of agent development. There are numerous independent efforts to create diverse agent frameworks optimized for specific tasks. Our work highlights a promising direction for leveraging these efforts effectively: a multi-agent orchestration layer that effectively harnesses diversity. Importantly, our orchestration layer operates at the agent framework level rather than within individual ones, demonstrating how such diversity of the entire community’s efforts can be systematically exploited to build more advanced intelligence systems.
>
> ### W2: more elaborate analysis and/or experimentation?
>
> Thanks for the suggestion. We have added more analysis and experimentation in the Appendix of the revision on questions including:
>
> - Which issues do the agents excel at? (**Appendix A.1.1**)
> - Why are they good at different issues? (**Appendix A.1.2**)
> - Can we learn a policy that chooses the agent in advance rather than selecting generated patches? (**Appendix A.1.3**)
> - Is the feedback from DEI helpful for refining the model generations instead of selecting them? (**Appendix A.2**)
> - When does DEI fail to provide correct feedback? (**Appendix A.3**)
>
> Since these additional analyses are also answers to some of your questions, we will explain them in detail in the response to those questions.

---

> ### Author Response · Authors · 2024-11-25
> **Response Part 2/3**
>
> ### Q1: Context in Eq. 3?
>
> We would like to clarify that the context is not fixed – it is sampled from the initial context distribution $c \sim \rho$. We can view the context distribution as the relevant repository information and issue description’s distribution in the training set. After sampling a context, your understanding is correct.
>
> ### Q2: How do we select the optimal policy for the given context?
>
> Yes. We compute the expected reward for every policy given the context and select the highest one. The implementation of the SWE-bench is shown in Step 3, where we use an LLM as the reward model to evaluate each policy’s expected reward (score).
>
> ### Q3 and Q5: Any learning involved in computing the DEI policy? Composed policies?
>
> Thank you for raising this insightful question! Our DEI policy formulation is designed to be general—it can either be a parameterized model trained on a dataset or, as you mentioned, an optimization program. In our current implementation for SWE-Bench, we adopt the latter approach, making it a zero-shot method that does not rely on any training data.
>
> Per your suggestion, we conducted an initial exploration of training a policy as a classifier that selects an agent for a given issue in advance. We report our findings in **Appendix A.1.3**. These classifiers are MLPs trained on the embeddings of issue descriptions. We are using voyage-code-2 embeddings for that purpose.
>
> We find that these **trained classifiers are not better than the random baseline** and were significantly outperformed by DeiBase, which leverages an LLM and incorporates both issue descriptions and generated patches into its decision-making process.
>
> We have added a discussion of these classifiers to the revision.
>
> | Model      | Input                   | Output       | Resolve% |
> | --------------- | ------------------------------------------ | ------------------- | -------- |
> | random baseline | -                     | which agent to use | 26.6%  |
> | classifier 1  | issue description             | which agent to use | 25.6%  |
> | classifier 2  | issue description + file structure     | which agent to use | 27.0%  |
> | DeiBase     | issue desc. + file structure + generations | which match is best | 33.3%  |
>
> We hypothesize that the reason behind this could be a lack of enough information, spurious features, lack of enough data (since we only have 150 issues in the training set), and embeddings not representing the issues well.
>
> Although the initial attempt proves unsuccessful due to the above contraints and the limited time, we recognize the potential of training a DEI policy using a pretrained LLM and a carefully curated dataset. Such a trained model could serve as a better judge model, dynamically composing and sequencing different policies’ partial solutions to address complex issues. While this represents an exciting and promising direction for future work, it is beyond the scope of this paper. We have noted this as an avenue for further exploration in our revised manuscript.

---

> ### Author Response · Authors · 2024-11-25
> **Response Part 3/3**
>
> ### Q4: Would it be possible to identify what the missing expertise is?
>
> Thank you for the suggestion. The expertise of agents does affect their performance, and DEI’s
>
> We have revised our manuscript to add analyses of which issues different open agents resolve and how their differences might be because of their abilities. We analyze the performance of the 4 open-source agents in DEI-Open, because we know their architecture and implementation.
>
> In summary, our findings (**Appendix A.1.1**) are:
>
> - The issues uniquely solved by only one agent are about 10% of all issues resolved by that agent
> - The issues resolved by OpenDevin are skewed towards the django repository.
> - OpenDevin is editing significantly more files and creating much longer patches (50x longer).
> - We do not observe a clear pattern correlating issue embeddings and agents’ ability to solve them.
> - Commonly resolved issues are generally better in description quality than those solved only by a subset of agents.
>
> In our qualitative analysis of why different agents resolve different issues (**Appendix A.1.2**), we identify the following potential reasons:
>
> - **Agents have different tool sets for bug localization.** Those using both string-matching tools and embedding-based retrieval have a better chance in finding where the bug is for some issues (sympy-23262 and django-15790).
> - **Bug reproduction and iterative testing is helpful in some cases.** OpenDevin tries to reproduce the bug described in an issue before resolving it, which might result in a test script that can be used to evaluate its own generation. While this creates much longer generations and costs more money, it also allows OpenDevin to resolve some issues that are not resolved by others, especially those with reproducible tests in their descriptions (sympy-15609 and sphinx-8435).
> - **Plausibility tests and mixing models.** Aider runs multiple iterations with different models – GPT-4o and Claude 3 Opus – when one iteration does not produce plausible patches. Plausibility is checked with a python linter and running pre-existing tests to make sure the changes do not break the system.
> - **Other ad hoc factors and spurious features, such as prompts.** While Aider and agentless are very similar in the way they localize the bugs (both prompt the LLM with a tree-like repository overview), there are many cases where one finds the bug and the other does not. For example, sympy-13971.
>
> However, these factors are entangled in their influence on the downstream performance and we do not see a clear pattern showing that DEI favors one expertise over another.
>
> Nonetheless, DEI’s feedback can also be used to refine model generations (**Appendix A.2**). We’ve tried this on the open-source agent *Agentless*. In particular, we experimented on 10 issues that were not resolved by it and got low scores from DEI. We modified the bug fixing part agentless framework to include DEI’s output, and refine the patches for at most 5 rounds. Once a patch gets more than 5 points from DEI, we stop refining it. With this refinement process, t**he number of fixed issues among the 10 went from 0 to 1, 2, 3, 5, 5 in the 5 rounds.**
>
> **Q6 and Q7: If you have as many members on the committee as you have different agents, wouldn't you always get the oracle answer? What is the minimal number of members on the committee to get an acceptable score? What is the cost ($) benefit (resolve %) trade-off?**
>
> Thank you for these insightful questions! To clarify, having as many members on the committee as there are agents does not guarantee the oracle answer because DEI does not always select the correct agent. The performance of the DEI committee can be constrained by the context window of the LLM used for scoring. As the number of agents increases, the length of the context (i.e., the combined input of all agent solutions) grows, which can degrade the model’s ability to accurately evaluate and select the best solution.
>
> We really appreciate the idea of the cost-benefit trade-off. For the closed-source agents in Figure 3a, we don’t know the cost of each agent run, but the number of agents can serve as an estimate for the cost. As Figure 3a suggests, 5 agents in the committee already yield a 33% resolve rate. Adding more agents to the committee still helps, but has a diminishing return (33% -> 35%).
>
> The same finding can be derived from the remaining 3 subplots in Figure 3. For the 3 open agents we evaluated, the number of agent runs is proportional to the cost (\\$). For moatless (which takes \\$0.6 for each instance), the cost to get to a decent performance plateau of 26% is \\$3.6 per instance. For agentless, which takes \\$0.4 for each instance, the cost to get to a decent performance plateau of 27% is \\$3.2 per instance.
>
> Thank you again for your insightful comments. We are glad to further discuss should you have more questions.

---

> > ### Comment · Reviewer_wcfY · 2024-11-26
> > **Author response**
> >
> > Thank you very much for the clarifications, I will raise my score.

---

> > > ### Author Response · Authors · 2024-11-27
> > > **Thanks for your feedback and updating the score**
> > >
> > > Dear Reviewer wcfY,
> > >
> > > Thank you for your encouraging feedback and recommendation of accepting the paper! Your valuable comments have greatly improved our results and presentation, making our contributions much clearer for the research community. Thank you very much!
> > >
> > > Best, Authors of submission 9222

---

### Official Review · Reviewer_nvyo · 2024-11-04

**Soundness:** 3
**Presentation:** 3
**Contribution:** 2
**Rating:** 6
**Confidence:** 4

**Summary:**

This paper introduces DEI (Diversity Empowered Intelligence), a framework designed to leverage the diverse capabilities of software engineering (SWE) agents. Through comprehensive analysis, the authors demonstrate that different SWE agents solve distinct sets of issues despite having similar overall performance metrics. The paper introduces DEI as a meta-module framework that harnesses this diversity through multi-stage rating and re-ranking, rather than attempting to build a single "best" agent. Their experimental results show significant improvements, with their best group achieving a 55% resolve rate on SWE-Bench Lite. The framework is well-documented with ablation studies and analysis.

**Strengths:**

- The paper is the first to investigate and establish that different SWE agents solve different types of issues.
- Proposes novel DEI meta-framework that uses diversity to build a better resolver than any single agent.
- Achieves 55% resolve rate on SWE-Lite beating baselines significantly.
- DEI-Base Open using open source LLMs achieves a 33% resolve rate outperforming many closed source LLMs.
- Well motivated with thorough analysis, and ablation studies.
- The release of code and prompts  makes the research reproducible.

**Weaknesses:**

- While the authors demonstrate that different SWE agents solve different sets of issues, they provide no analysis of why this occurs (e.g., no investigation of which types of issues each agent excels at, or how agent architectures relate to their success patterns). This limits understanding of the relationship between agent design and issue-solving capabilities.

- The framework uses fixed agent groups (DEIBASE-1/2/Open) where all agents in a set contribute patches that are then scored. There is no learning mechanism to dynamically select agents based on past performance or issue characteristics. As a result, the system relies entirely on inherent diversity and fails to leverage agent specialization.

- DEI employs a fixed sequential review process (issue -> patches -> scoring) without iterative refinement or feedback loops between components. This linear approach potentially misses opportunities for improved patch selection that could be achieved through multi-round evaluation.

**Questions:**

- Could you provide a deeper analysis of why different agents excel at different issues? For example, have you analyzed patterns in the types of issues each agent solves successfully or investigated how specific agent architectures (e.g., execution vs non-execution-based agents) relate to their issue-solving capabilities?
- Could DEI's architecture be extended to provide feedback to the agents about why their patches failed, potentially enabling them to generate better patches?
- Could you provide a detailed analysis of DEI's failure cases? For instance, are there specific types of patches or issues where the scoring mechanism consistently makes wrong decisions?

---

> ### Author Response · Authors · 2024-11-25
> **Response Part 1/2**
>
> Dear Reviewer nvyo,
>
> We sincerely thank you for the time dedicated to reviewing our paper and the helpful comments. Here’s our response to your comments and questions.
>
> ### W1: Which issues do the agents excel at? Why are they good at different issues?
>
> In addition to the grids in Figure 1a showing issues resolved by different agents, we have added a detailed analysis in **Appendix A.1** of the revised manuscript. We analyze the performance of the 4 open-source agents in DEI-Open, because we know their architecture and implementation.
>
> In summary, our findings are:
>
> - The issues uniquely solved by only one agent are about 10% of all issues resolved by that agent
> - The issues resolved by OpenDevin are skewed towards the django repository.
> - OpenDevin is editing significantly more files and creating much longer patches (50x longer).
> - We do not observe a clear pattern correlating issue embeddings and agents’ ability to solve them.
> - Commonly resolved issues are generally better in description quality than those solved only by a subset of agents.
>
> In our qualitative analysis of why different agents resolve different issues (**Appendix A.1.2**), we identify the following potential reasons:
>
> - **Agents have different tool sets for bug localization.** Those using both string-matching tools and embedding-based retrieval have a better chance in finding where the bug is for some issues (sympy-23262 and django-15790).
> - **Bug reproduction and iterative testing is helpful in some cases.** OpenDevin tries to reproduce the bug described in an issue before resolving it, which might result in a test script that can be used to evaluate its own generation. While this creates much longer generations and costs more money, it also allows OpenDevin to resolve some issues that are not resolved by others, especially those with reproducible tests in their descriptions (sympy-15609 and sphinx-8435).
> - **Plausibility tests and mixing models.** Aider runs multiple iterations with different models – GPT-4o and Claude 3 Opus – when one iteration does not produce plausible patches. Plausibility is checked with a python linter and running pre-existing tests to make sure the changes do not break the system.
> - **Other ad hoc factors and spurious features, such as prompts.** While Aider and agentless are very similar in the way they localize the bugs (both prompt the LLM with a tree-like repository overview), there are many cases where one finds the bug and the other does not. For example, sympy-13971.
>
> ### W2: Can we learn which agent to use for each issue?
>
> We tried training classifiers and report the results in the following table (also added to the revision). These classifiers are MLPs trained on the embeddings of issue descriptions. We are using voyage-code-2 embeddings for that purpose.
>
> We find that these **trained classifiers are not better than the random baseline**, and they are significantly worse than DeiBase, which takes both the issue description and generated patches as inputs.
>
> We hypothesize that the reason behind this could be a lack of enough information, spurious features, lack of enough data (since we only have 150 issues in the training set), and embeddings not representing the issues well.
>
> In light of your suggestion, we have added discussion of these classifiers to the revision.
>
> | Model      | Input                   | Output       | Resolve% |
> | --------------- | ------------------------------------------ | ------------------- | -------- |
> | random baseline | -                     | which agent to use | 26.6%  |
> | classifier 1  | issue description             | which agent to use | 25.6%  |
> | classifier 2  | issue description + file structure     | which agent to use | 27.0%  |
> | DeiBase     | issue desc. + file structure + generations | which match is best | 33.3%  |
>
> ### W3: Can DEI feedback be used for patch refinement?
>
> We are glad you brought this up.
>
> Yes, we’ve tried this on the open-source agent *Agentless*. In particular, we experimented on 10 issues that were not resolved by it and got low scores from DEI. We modified the bug fixing part agentless framework to include DEI’s output, and refine the patches for at most 5 rounds. Once a patch gets more than 5 points from DEI, we stop refining it. With this refinement process, t**he number of fixed issues among the 10 went from 0 to 1, 2, 3, 5, 5 in the 5 rounds.**
>
> This experiment shows that DEI feedback can indeed be used for patch refinement. We appreciate your suggestion and will include a larger-scale experiment in our revision.
>
> ### Q1: Could you provide a deeper analysis of why different agents excel at different issues?
>
> Please see our response to W1 and updated **Appendix A.1**.
>
> ### Q2: Could DEI's architecture be extended to provide feedback to the agents?
>
>
> Please see our response to W3 and updated **Appendix A.1**.

---

> ### Author Response · Authors · 2024-11-25
> **Response Part 2/2**
>
> ### Q3: Could you provide a detailed analysis of DEI's failure cases?
>
> Thanks for your suggestion.
>
> DEI follows our rubric to score the patches. Each stage in its analysis corresponds to some points in the scoring rubric, therefore needs to be analyzed separately.
>
> We analyzed 20 of 35 failure cases of DEI in DEI-Open by manually annotating the stages (location explanation, patch explanation, conflict detection) where DEI failed to make the correct decision.
>
> For each case, we analyze for one false positive patch and one false negative patch. Here’s the result.
>
> |                | Location Explanation | Patch Explanation | Conflict Detection |
> | -------------- | -------------------- | ----------------- | ------------------ |
> | False Positive | 5                    | 10                | 7                  |
> | False Negative | 4                    | 2                 | 15                 |
>
> Note that each row can sum up to more than 20, because there can be multiple stages where DEI makes mistakes.
>
> From the table above, we find that DEI tends to be misled during the patch explanation stage of an incorrect patch. For a correct patch, it tends to mistakenly “find” conflicts with the existing code.
>
> Fewer errors are made during the location explanation stage. This indicates that DEI is better at telling if the patch is modifying the correct file.
>
> We really appreciate your efforts in helping us refine our paper. Should you have any further questions, please do not hesitate to let us know.

---

> > ### Comment · Reviewer_nvyo · 2024-11-27
> >
> > Thanks for the response, I've increased my score!

---

> > > ### Author Response · Authors · 2024-11-27
> > > **Thanks for raising the score**
> > >
> > > Dear Reviewer nvyo,
> > >
> > > Thanks for your recognition of our efforts and updating the score. We are very delighted to hear that your concerns have been properly addressed.
> > >
> > > Best, Authors of Submission 9222

---

### Meta-Review · Area_Chair_Y4iU · 2024-12-23

**Metareview:**

An ensemble method for software engineering. Interesting new approach to an important problem with very good results on an important benchmark. Clear accept.

The diversity approach makes me wonder whether have looked at quality-diversity methods such as MAP-Elites. QD methods have been used for ensemble classification before, and also separately together with LLMs for code generation, e.g. LLMatic by Nasir et al.

PS: Fun acronym, and having a box labeled "DEI committee review" in your system diagram is hilarious :)

**Additional Comments On Reviewer Discussion:**

Productive and thorough reviewer discussion.

---

### Decision · Program_Chairs · 2025-01-22

Accept (Poster)